# Tuning myosin-driven sorting on cellular actin networks

**Rizal F Hariadi[1], Ruth F Sommese[1], Sivaraj Sivaramakrishnan[1,2,3]***

[1]Department of Cell and Developmental Biology, University of Michigan, Ann Arbor, United States; [2]Department of Biophysics, University of Michigan, Ann Arbor, United States; [3]Department of Biomedical Engineering, University of Michigan, Ann Arbor, United States

**Abstract** Myosin V and VI are antagonistic motors that cohabit membrane vesicles in cells. A systematic study of their collective function, however, is lacking and forms the focus of this study. We functionally reconstitute a two-dimensional actin-myosin interface using myosin V and VI precisely patterned on DNA nanostructures, in combination with a model keratocyte actin meshwork. While scaffolds display solely unidirectional movement, their directional flux is modulated by both actin architecture and the structural properties of the myosin lever arm. This directional flux can be finely-tuned by the relative number of myosin V and VI motors on each scaffold. Pairing computation with experimental observations suggests that the ratio of motor stall forces is a key determinant of the observed competitive outcomes. Overall, our study demonstrates an elegant mechanism for sorting of membrane cargo using equally matched antagonistic motors, simply by modulating the relative number of engagement sites for each motor type.

## Introduction

Membrane sorting in the secretory and endocytic pathways occurs in the midst of the actin cytoskeleton, and involves a range of unconventional myosins that link membrane components to the actin network (*Hartman et al., 2011*). However, traditional reconstituted systems to study membrane traffic do not incorporate the effects of actin-myosin interactions (*Lee et al., 2004*; *Zanetti et al., 2012*). Additionally, while unconventional myosins are necessary for timely membrane traffic, their functional role is not apparent in live cell studies (*Hasson et al., 1997*; *Sahlender et al., 2005*; *Hartman et al., 2011*). The bulk of our knowledge of unconventional myosin function instead stems from single molecule biophysical and structural studies, which demonstrate distinct functional regimes for actin-myosin interactions including bi-directional motion, unidirectional transport, and mechano-sensitive anchoring (*Trybus, 2008*; *Spudich and Sivaramakrishnan, 2010*). There remains, however, a considerable gap between the insights gained from single-motor studies and a mechanistic understanding of cargo transport in living cells. Furthermore, membrane trafficking often involves multiple disparate motor types, and their collective function cannot be trivially extrapolated from single molecule studies. In this study, we focus on myosin V and VI, two opposing unconventional myosins that co-reside on membrane vesicles in neuronal growth cones (*Suter et al., 2000*). Myosin V has been implicated in secretory traffic, whereas myosin VI facilitates timely endocytosis (*Suter et al., 2000*; *Kneussel and Wagner, 2013*). Individual myosin V and VI molecules within a transport ensemble may coordinate, cooperate, or mechanically impede one another to influence collective movement (*Rogers et al., 2009*; *Sivaramakrishnan and Spudich, 2009*; *Lu et al., 2012*). Hence, studies with mixed motor ensembles are essential to define the function of myosins in membrane trafficking.

All myosins share a conserved catalytic domain that converts the chemical energy of ATP hydrolysis into a unidirectional mechanical stroke of the motor lever arm. In the case of myosin V and VI, they are

*For correspondence: sivaraj@umich.edu

**Competing interests:** The authors declare that no competing interests exist.

**eLife digest** Proteins and other molecules can be moved around a cell within bubble-like compartments called vesicles. These vesicles can travel along filaments made of a protein called actin, which forms a network that criss-crosses the cell. A family of motor proteins called myosin bind to the vesicles and are responsible for pulling them along the actin filaments. For example, myosin V pulls vesicles towards the 'plus-end' of the filament or the outer edges of the cell, while myosin VI pulls them in the opposite direction towards the 'minus-end' or the interior of the cell.

Both proteins are often found on the same vesicle, and it is not clear in which direction such a vesicle will move. Hariadi et al. have shed new light on this question by sticking different combinations of myosin V and myosin VI proteins to a tiny nanostructure made of DNA and using a microscope to watch it move on actin.

When a nanostructure with one myosin V and one myosin VI protein was placed on a single actin filament, it moved towards the plus-end of the filament. However, when it was placed on a two-dimensional network of actin filaments, the nanostructure was equally likely to move in either direction. Therefore, the architecture of the actin filaments influences the outcome of the competition between the two motor proteins.

When both types of myosin protein were present, the nanostructure was pulled along the filament more slowly than when only one type was present. This suggests that myosin V and myosin VI are involved in a 'tug of war' on the actin filament. Next, Hariadi et al. altered the numbers of myosin V and myosin VI proteins on the nanostructure. The direction in which the nanostructure moved depended on the ratio of motor proteins present: when there were more myosin V proteins than myosin VI proteins, the nanostructure moved towards the plus-end, and vice versa.

Hariadi et al.'s findings suggest that cells direct the movement of vesicles around a cell by altering the relative number of myosin V and myosin VI proteins bound to each vesicle.

considered evenly matched antagonistic motors (*Trybus, 2008*; *Spudich and Sivaramakrishnan, 2010*). Both motors are thought to bind membrane cargo as dimers; myosin V through a coiled-coil motif following its lever arm that natively homodimerizes it, and myosin VI presumably through dimeric adaptor proteins that link it to cargo (*Mehta et al., 1999*; *Buss and Kendrick-Jones, 2008*). Homodimers of either myosin move processively on actin filaments with similar step sizes (V—36 nm; VI—30 nm), stepping kinetics (V –12 s$^{-1}$; VI –9 s$^{-1}$), and stall forces (V ~3 pN; VI ~2 pN) albeit in opposing directions (*Mehta et al., 1999*; *Rief et al., 2000*; *Rock et al., 2001*; *Nishikawa et al., 2002*; *Yildiz et al., 2003*; *Altman et al., 2004*; *Uemura et al., 2004*). All myosin levers, with the exception of myosin VI, swing towards the barbed (plus) end of the actin filament. In the case of myosin VI, a unique insert reverses the direction of its lever stroke towards the pointed (minus) end of the actin filament (*Liao et al., 2009*; *Spudich and Sivaramakrishnan, 2010*). With the plus-ends of actin networks oriented toward the cell periphery, plus-end directed myosin V thus contributes to exocytosis, whereas minus-end directed myosin VI is critical to endocytosis (*Hartman et al., 2011*). Finally, despite their many similarities, myosin V and VI have structurally distinct lever arms. The myosin V lever consists of six light chain binding IQ-motifs wrapped with calmodulin light chains (*Trybus, 2008*). The myosin VI lever is composed of two calmodulin-binding IQ-motifs followed by a pliable proximal tail domain, and a semi-rigid single α-helical domain (*Spudich and Sivaramakrishnan, 2010*).

Translating the detailed structural understanding of individual myosin V and VI into cellular function, specifically when they cohabit the same scaffold, remains an outstanding challenge. *Ali et al. (2011)* reported that tethering a single myosin V and a single VI homodimer on a quantum dot leads to unidirectional motion on single actin filaments, with myosin V dominating the competition (79% of processive runs towards the plus-end of actin filaments). We recently extended this finding to DNA nanostructures containing two myosin V and two myosin VI molecules interacting with a keratocyte-derived actin network (*Hariadi et al., 2014*). While we did observe solely unidirectional movement, in contrast to *Ali et al. (2011)*, myosin V and VI were evenly matched in our system (52% of processive runs towards the keratocyte cell periphery). Our previous study focused on trajectory shapes and did not address this observed discrepancy in the outcome of the competition. Further, the generality of these observations for

different ratios of myosin V and VI and the mechanisms that control directionality remain unexplored and form the focus of this study.

Here, we use DNA nanotechnology to precisely scaffold defined collections of myosin V and VI and pair it with both single actin filaments and a model cellular actin network derived from the extensive lamellipodium of fish epidermal keratocytes (*Hariadi et al., 2014*). Consistent with previous reports (*Ali et al., 2011*; *Hariadi et al., 2014*), we observe solely unidirectional movement regardless of actin architecture or relative myosin number. However, for matched scaffolds we find that the directional flux is dependent on both actin architecture and the structural properties of the myosin lever arm. This directional flux is finely-tuned by the relative number of myosin V and VI motors on each scaffold. By pairing computation and experiment, we identify a single mechanical parameter that defines regimes in any motor ensemble wherein this mechanism is likely to be observed. Overall, our study demonstrates an elegant mechanism for sorting of membrane cargo simply by modulating the relative number of engagement sites for each motor type. For matched, but opposing motors such as myosin V and VI, this mechanism is necessary and sufficient to precisely control sorting of tethered scaffolds.

## Results

### Combining DNA nanostructures with defined motor composition and 1D/2D actin tracks

To investigate the role of actin organization in trafficking, DNA nanostructures containing a defined number of antagonistic myosins (V and VI; *Figure 1A*; *Figure 1—figure supplements 1, 2*; *Supplementary file 1*) were examined on two distinct actin architectures, namely one-dimensional actin filaments (*Figure 1B,D,F*) and dense two-dimensional actin networks (*Svitkina and Borisy, 1998*; *Schaus et al., 2007*) (*Figure 1C,E,G*). Precise positioning of myosin V and VI on the origami scaffold was achieved using myosins labeled with single-stranded DNA oligonucleotides complementary to attachment sequences projecting from the scaffold strand (1–6 per scaffold; *Figure 3—figure supplement 1*). DNA nanostructures with varying numbers of myosin are denoted as $x$V:$y$VI, where '$x$' is the number of myosin V dimers and '$y$' is the number of myosin VI dimers per scaffold. For the 2D actin networks, we used detergent-extracted keratocytes (*Hariadi et al., 2014*) (*Figure 1C*), which have a sufficiently large surface area (~10 μm × ~30 μm) allowing for simultaneous tracking of multiple myosin-labeled scaffolds. Experiments involving 1D actin filaments provide a confined set of actin-myosin interactions, with each myosin having either a forward (red rectangle) or a backward (gray rectangle) binding site available (*Figure 1F*). The 2D actin networks, on the other hand, provide a more complex energy landscape for the myosins to navigate, as there are multiple binding sites for both forward (red arc) and backward (gray arc) steps (*Figure 1G*).

### Actin architecture influences competitive outcome

Two previous reports suggest that equal numbers of myosin V and VI anchored to the same scaffold display solely unidirectional movement (*Ali et al., 2011*; *Hariadi et al., 2014*). However, they disagree in the observed outcome of the competition. Myosin V dominates the competition (79%) when it is tethered to myosin VI through a quantum dot (2 total) and the two compete on a single actin filament (*Ali et al., 2011*). In contrast, myosin V and VI are evenly matched (myosin V wins 52%) when two of each motor (4 total) are tethered to a DNA nanostructure and they compete on a two-dimensional cellular actin network. This discrepancy between the observations could stem from either the scaffold type (quantum dot vs DNA nanostructure), the total motor number (2 vs 4), or the actin architecture (single filament vs keratocyte-derived actin network). We first tested the influence of scaffold type by assessing the competition between a single myosin V dimer and a single myosin VI dimer on 1D actin filaments (*Figure 2*). In positive controls, ØV:2VI scaffolds (*Figure 2A*) move toward the minus-end of the actin filaments, whereas 2V:ØVI scaffolds (*Figure 2C*) travel toward the plus-end. Consistent with previous reports (*Ali et al., 2011*; *Hariadi et al., 2014*), scaffolds with both myosin V and myosin VI (1V:1VI) commit to a single direction on actin filaments (>99%; *Figure 2B*) with no directional reversal detected. The movement of 1V:1VI scaffolds on single actin filaments is dominated by plus-end directed movement ($\Phi_{out}$ = 68 ± 1%; *Figure 2E*), which is qualitatively consistent with previous observations using quantum dot scaffolds (79% plus-end directed [*Ali et al., 2011*]).

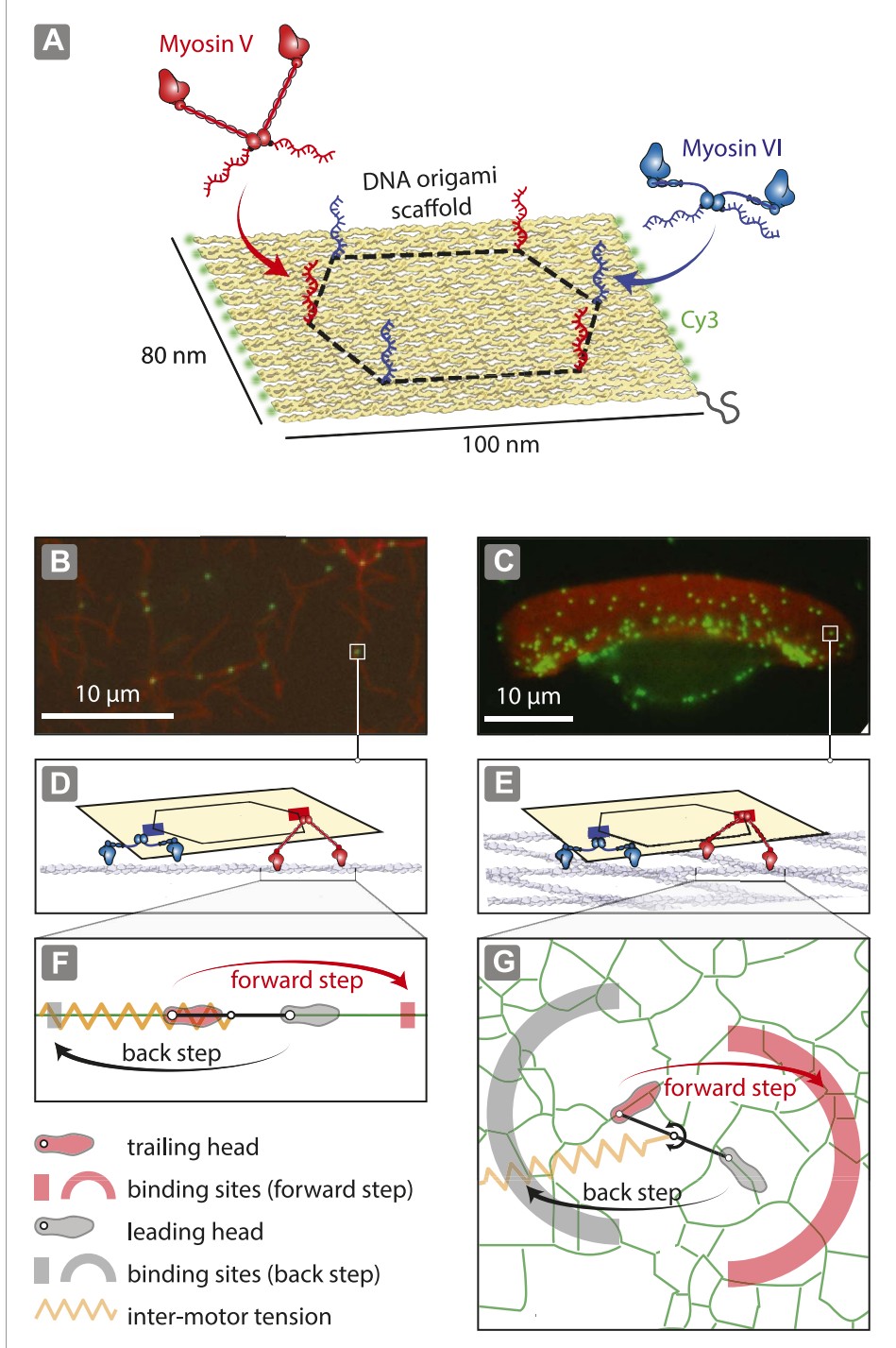

**Figure 1**. Reconstitution of myosin-driven cargo sorting on 1D and 2D actin tracks. (**A**) Illustration of a programmable DNA scaffold (*Rothemund, 2006*) with six attachment sites at the vertices of a hexagon (dashed-line, 35-nm side), yielding 122 unique myosin V and VI combinations. Myosin V and VI were engineered with SNAP tags (*Hariadi et al., 2014*) for covalent attachment of unique DNA oligonucleotides. These DNA oligonucleotides hybridize with complementary sequences extending from the scaffold. (**B–C**) Representative snapshot of scaffold-motor complexes (green) on actin filaments (**B**) and a keratocyte actin network (**C**). Actin was stabilized and labeled with Alexa488-phalloidin (red). (**D–E**) Schematics depicting the interaction of scaffolds (yellow) with 1 myosin V (red) and 1 myosin VI (blue) on an actin filament (**D**) and on the surface of the keratocyte actin network (**E**). The motors and actin tracks are drawn approximately to scale. The keratocyte actin network is depicted by actin filaments oriented at ±35°, which

*Figure 1. continued on next page*

*Figure 1. Continued*

corresponds to the characteristic Arp2/3 branch angle (*Maly and Borisy, 2001*). Mesh size of the keratocyte actin network (~30 nm) (*Svitkina et al., 1995*) is comparable to the step size of myosin V (~35 nm) and VI (~30 nm) (*Rock et al., 2001*; *Yildiz et al., 2003*). (**F–G**) Hand-over-hand model of dimeric myosin stepping on 1D (**F**) and 2D (**G**) actin tracks. The competition between antagonistic myosins gives rise to inter-motor tension depicted as a simple harmonic spring (orange). For inter-motor tension below the stall force, the trailing head (light red) moves 36 nm forward (red arrow) to a new position within the forward-step target zone (shaded red areas), while the leading head (gray) remains stationary. High inter-motor tension induces a backward step (black arrow) of the leading head to a target site within the back-step target zone (shaded gray areas).

The following figure supplements are available for figure 1:

**Figure supplement 1**. Flat rectangular DNA origami scaffold.

**Figure supplement 2**. Sequence diagram for a flat rectangular DNA origami scaffold.

Hence, scaffold type (quantum dot vs DNA nanostructure) is not the key determinant of competitive outcome. We next examined the influence of actin architecture. In contrast to single actin filaments, both plus and minus-end directed movement is equally represented ($\Phi_{out} = 52 \pm 1\%$; *Figure 2E*) for 1V:1VI scaffolds moving along 2D keratocyte actin networks. Hence, the discrepancy between previous reports using quantum dots (*Ali et al., 2011*) and DNA nanostructures (*Hariadi et al., 2014*) stems primarily from the actin architecture.

## Directional flux of scaffolds is linearly dependent on relative number of myosin V and VI

In order to assess the role of relative motor number on competitive outcome, we next tested scaffolds with varying ratios of myosin V and myosin VI motors ($x$V:$y$VI; *Figure 3—figure supplement 1*) on 2D actin networks (*Figure 3A*). In every combination, the origami scaffold commits to a single direction, either towards the cell periphery or the cell center (*Figure 3B*). The relative number of scaffolds that move to the cell center and cell periphery ($\Phi_{out}$ or $\Phi_{in}$), however, varies linearly with the fraction of myosin V or myosin VI (*Figure 3B–D*). Thus, while the scaffolds have a dedicated direction of movement on both 1D and 2D actin landscapes, the underlying competition (tug-of-war) systematically influences the directional flux.

## Engagement of antagonistic motors with the underlying actin network

The speed of nanostructures (1V:1VI = (+) 162 ± 7 nm/s; (−) 66 ± 5 nm/s) along actin filaments is significantly slower than nanostructures containing only two myosin V (2V:ØVI = (+) 273 ± 8 nm/s) or two myosin VI (ØV:2VI = (−) 130 ± 7 nm/s) (*Figure 2D*). Likewise, the speed of nanostructure movement on the keratocyte network decreases as the difference in the number of the two motor types approaches zero (*Figure 3C*). These reductions in speed with antagonistic motors are in agreement with the previously published experiments involving quantum dots conjugated to one myosin V and one myosin VI (*Ali et al., 2011*). Based on the reduction in speed for antagonistic ensembles, as compared to groups of one myosin type, we hypothesized that all of the motors can continuously interact with the actin tracks and collectively engage in competition. To test this hypothesis, scaffolds were formed with three myosin V and three myosin VI (3V:3VI), where one of the motor types was attached by photo-cleavable linkers (*Figure 4A–B*). Regardless of which motor type is cleaved, removal of one myosin type from the competition increases the speed and results in a single direction of movement (*Figure 4C–F*). The directional switch and increase in speed after photo-cleavage indicate that all motors, regardless of type, are able to access the actin tracks and engage in continuous competition. Together, these observations suggest that the collective movement is due to a continuous interaction of both motor types, and not due to detachment of losing motors from the actin track (or scaffold), when overpowered by the winning motor. Lastly, the underlying continuous interaction is also consistent with our previous observation that myosin V changes the trajectory shape of ensembles of myosin V and VI on 2D actin tracks (*Hariadi et al., 2014*).

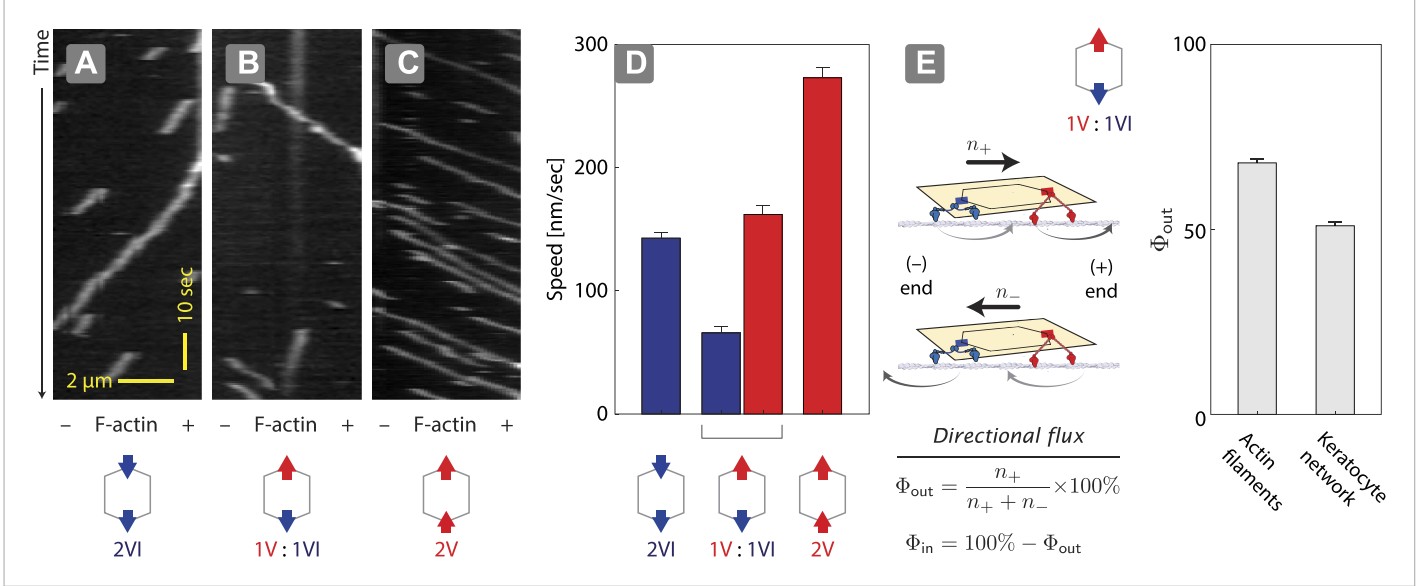

**Figure 2.** Unidirectional movement and sorting of scaffolds with myosin V and VI along single actin filaments. (**A–C**) Kymographs showing the movement of indicated motor ensembles along actin filaments. Scaffolds with myosin V and VI display unidirectional movements toward plus-or minus-ends of the actin filament. The gray hexagon represents the organization of attachment sites on the scaffold, the red and blue arrows denote myosin V and VI, respectively. (**D**) Speed of plus-end (blue) and minus-end (red) directed movement of indicated scaffolds on actin filaments. Error bars are S.E.M. (**E**) Relative frequency of plus-end ($n_+$) and minus-end ($n_-$) directed movement for 1V:1VI scaffolds on actin filaments and keratocyte actin networks. Outward flux ($\Phi_{out}$) is defined as the fraction of plus-end directed trajectories. Error bars are S.E.M. and were generated by bootstrapping ($N \geq 202$ trajectories; $\geq 3$ experiments).

## Stochastic simulations identify key parameters that drive unidirectional movement

To gain insight into the structural mechanisms of the observed directional flux, a minimal stochastic simulation was used to model the contributions of inter-motor tension and intra-motor strain to the competition between opposing motors (*Figure 5*; 'Materials and methods'). In the model, two opposing motors are coupled mechanically through a linear spring of strength $k_s$ (*Figure 5A*). Since the motor proteins are the most flexible components of the scaffold–motor complex, $k_s$ is dominated by the flexibility of the myosin motors. Each motor consists of two catalytic heads that are connected by a lever arm with flexural rigidity $k_F$. Each motor also has a comparable, albeit mismatched, stall force ($1 \leq F_{high}/F_{low} \leq 2$), where $F_{high}$ and $F_{low}$ are the stall forces of the stronger (myosin V) (*Mehta et al., 1999*; *Uemura et al., 2004*) and weaker (myosin VI) (*Rock et al., 2001*; *Nishikawa et al., 2002*; *Altman et al., 2004*) motors, respectively. Our model assumes that a motor can only perform a forward step if the resulting inter-motor tension ($T$) is less than its stall force (*Figure 5A*). A successful step increases the inter-motor tension by $\Delta T = k_s \bullet s$, where $s$ is the motor step size. Initially $T$ is set to zero and both motors take forward steps stochastically in opposite directions, increasing $T$ with each step. This sequence of movement proceeds until a forward step increases $T$ beyond the stall force of the stepping motor, which undergoes a conformational change that leads to its preferential back-stepping (*Gebhardt et al., 2006*) thereby relieving inter-motor tension (*Ali et al., 2011*).

Stochastic simulations that follow this model lead to solely unidirectional movement, with the relative number of plus ($n_+$) and minus ($n_-$) end directed scaffolds dependent on the normalized inter-motor tension per step ($\Delta T/F_{low}$) and stall force ratio ($r_s = F_{high}/F_{low}$) (*Figure 5B*). For equally matched motors ($r_s = 1$), there is an equal probability of trajectories moving in either direction ($\Phi_{out} = 50\%$). For $1 < r_s < 2$, the model shows that $\Phi_{out}$ can be tuned from 50% to 100% depending on the value of $\Delta T/F_{low}$ (*Figure 5B*). For $\Delta T/F_{low} < 0.5$, the inter-motor tension exceeds the stall force of the weaker motor, with the stronger motor winning most of the competitions ($\Phi_{out} > 80\%$). However, for $0.5 < \Delta T/F_{low} < 1$ and stochastic stepping, there is a finite and increasing probability of

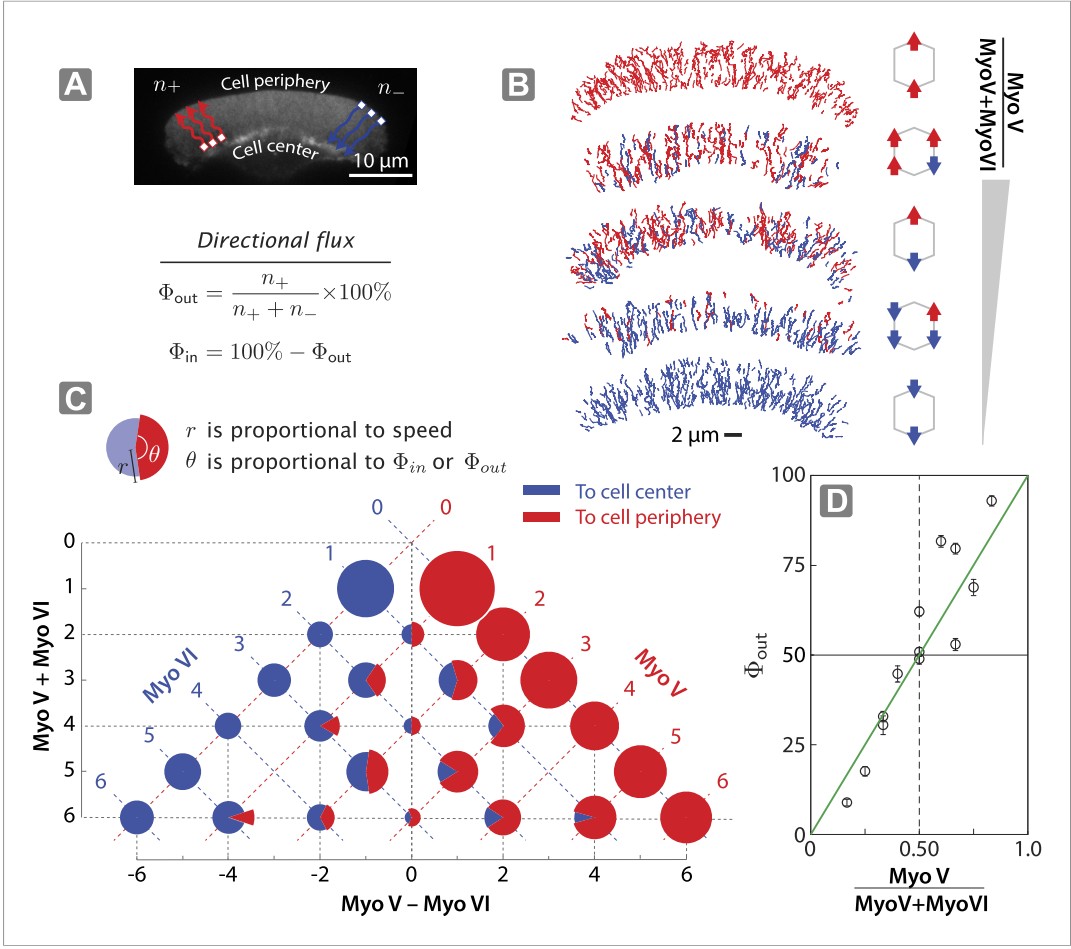

**Figure 3**. Scaffolds show unidirectional movement along actin networks with directional flux controlled by relative motor number. (**A**) Schematic of scaffold-myosin movement on the surface of the keratocyte actin network. Inward ($\Phi_{in}$) and outward directional flux ($\Phi_{out}$) are calculated as the fraction of myosin scaffolds moving towards the cell center and periphery, respectively. (**B**) Sample trajectories of scaffolds on keratocyte actin networks with movement towards the cell center in blue and the cell periphery in red. (**C**) Summary plot depicting influence of relative motor number on directionality (red and blue) and speed (radius). The plot is aligned such that the y-axis represents total motor number (myosin V + myosin VI), whereas the x-axis represents the difference between the two myosin types (myosin V − myosin VI). Red or blue dashed lines correspond to scaffolds with equal numbers of myosin V or VI motors, respectively. (**D**) Outward flux ($\Phi_{out}$) varies linearly with the difference between the number of myosin V and VI (green line, R = 0.80). Positive and negative values indicate net movement towards cell periphery and cell center, respectively. Error bars are S.E.M. and were generated by bootstrapping (N = 58–1897 trajectories; 3–4 keratocytes).

The following figure supplement is available for figure 3:

**Figure supplement 1**. Scaffolds precisely patterned with myosin V and/or VI.

---

inter-motor tension exceeding the stall force of the stronger motor ('Materials and methods'), resulting in the weaker motor winning the competition (60% < $\Phi_{out}$ < 80%). This regime captures the experimentally measured $\Phi_{out}$ of 68% (⊗; *Figure 5B*), given the previously reported stall forces of myosin V (*Mehta et al., 1999*; *Uemura et al., 2004*) and VI (*Rock et al., 2001*; *Nishikawa et al., 2002*; *Altman et al., 2004*).

In our model for movement on a 1D actin filament track, the $\Delta T$ for myosins with equal step sizes is the same regardless of which motor steps forward ($\Delta T = k_s \bullet s$). Parallel simulations on digitized keratocyte actin networks, however, incorporate an additional parameter, namely flexural rigidity of the myosin lever arm $k_F$ (*Hariadi et al., 2014*) (*Figure 5C* and *Figure 5—figure supplement 1*), to account for the misalignment of the lever arm relative to the local actin filament where the myosin

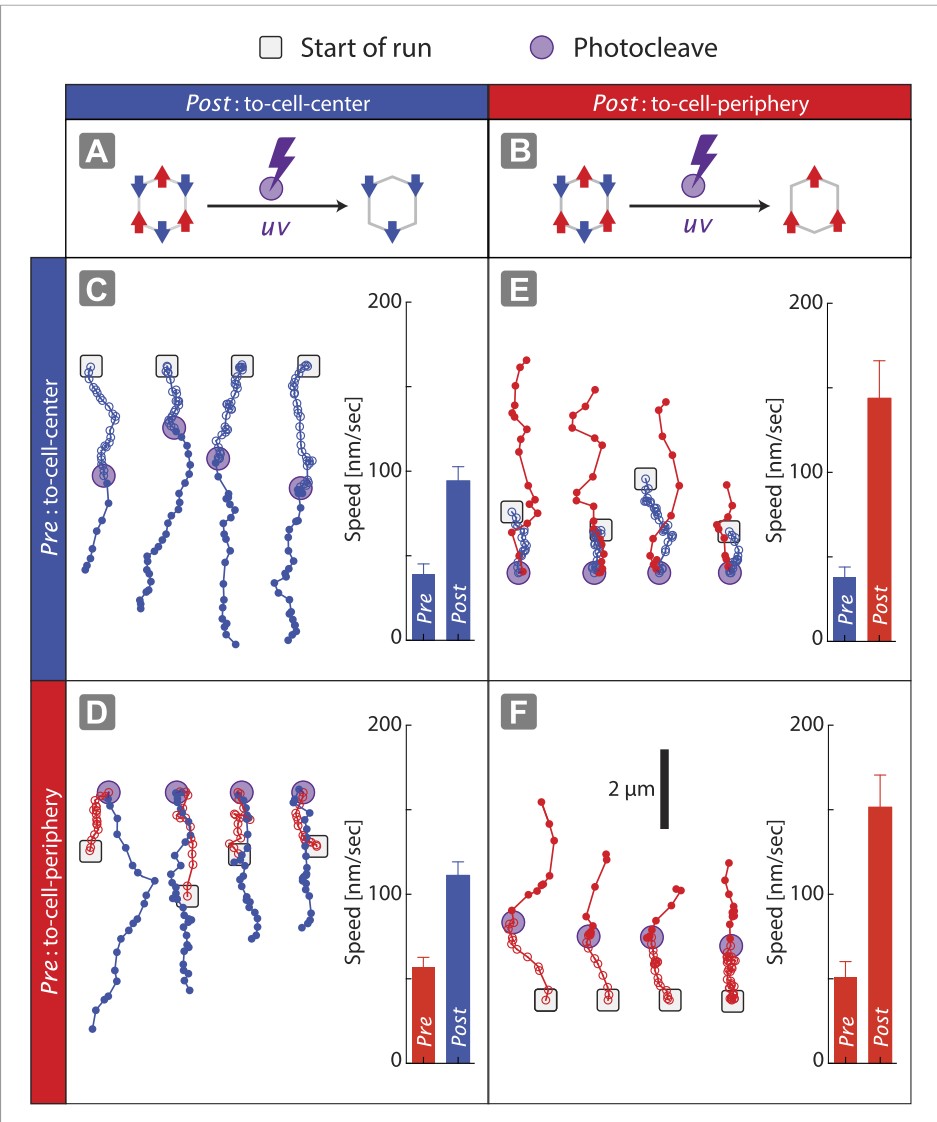

**Figure 4.** Disengagement of one motor species resolves the competition. (**A–B**) Schematics of mixed-motor scaffolds (gray hexagons) with three myosin V (**A**; red arrows) or three myosin VI (**B**; blue arrows) attached by photo-cleavable linkers. UV-induced photo-cleavage (purple lightning bolt) converts mixed-motor scaffolds to scaffolds with only myosin VI (**A**) or only myosin V (**B**). (**C–F**) Representative scaffold trajectories for photo-cleavable experiments on keratocyte networks and their corresponding mean speed. Black squares mark the start of the trajectories, and purple circles indicate the start of photo-cleavage. Individual data points in each trajectory, before and after photo-cleavage, are indicated as open or closed circles, respectively. Disengagement of myosin V (**C** and **D**) or myosin VI (**E** and **F**) results in movement toward cell-center or cell-periphery, respectively. In all cases, photo-cleavage leads to significant increase in speed (p < 0.01). Error bars are S.E.M. (N ≥ 19 trajectories; ≥ 5 keratocytes).

head is bound (*Figure 5* and *Figure 5—figure supplement 1L*). Simulations on these networks show that the mean inter-motor tension per step (Δ*T*) can be significantly influenced by relative torsional stiffness ($k_F/k_s$), regardless of network mesh size or inter-motor stiffness (*Figure 5C–D* and *Figure 5—figure supplements 1–3*). On a 2D network, the higher the $k_F/k_s$ of a motor, the greater the Δ*T* when it steps forward. Thus one can model movement along a 2D network with a similar simulation on a 1D track by redefining Δ*T* such that different Δ*T* values are accrued in each step based on which motor steps forward (Δ$T_{high}$ and Δ$T_{low}$ for rigid and flexible motors respectively). For such simulations on 2D networks, linking a rigid motor ($k_F/k_s \gg 1$; Δ$T_{high}$) to a more flexible one ($k_F/k_s \ll 1$; Δ$T_{low}$) is sufficient to systematically bias the competition in favor of the flexible

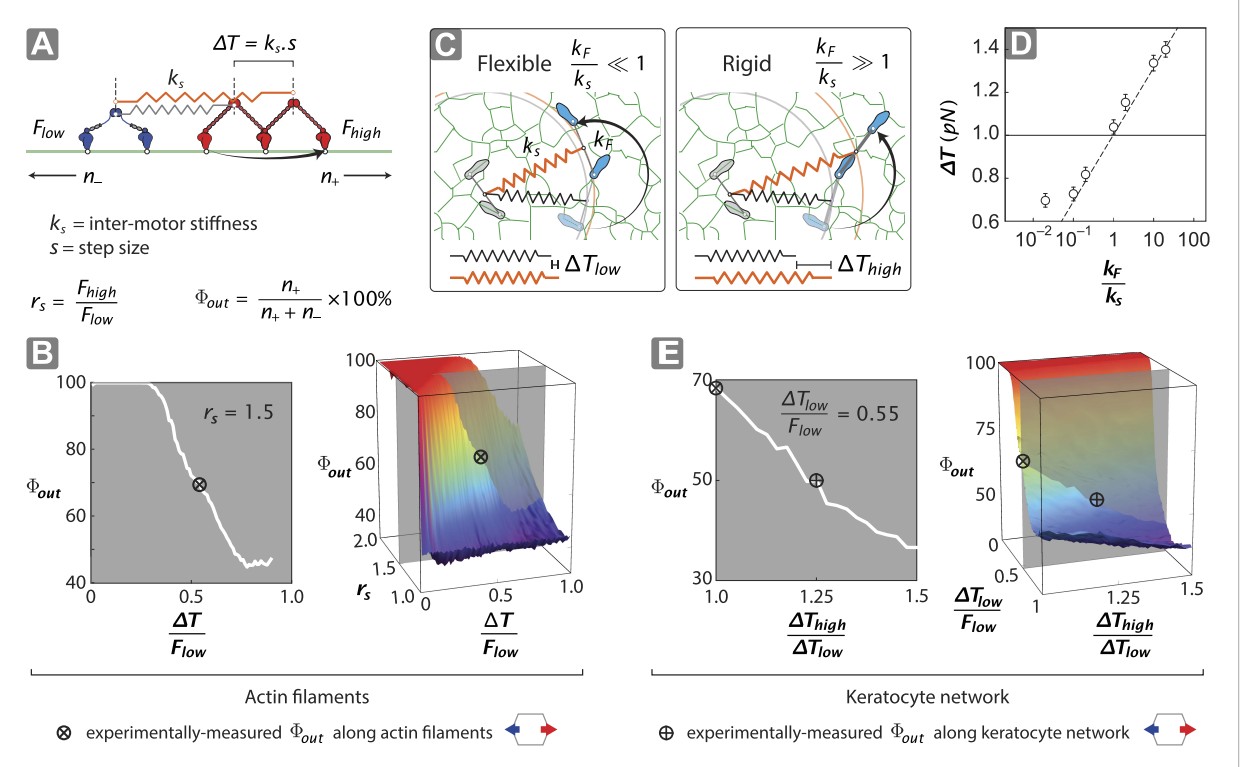

Figure 5. Stall force ratio, actin architecture, and myosin rigidity together tune directional flux. (A) Minimal model of coupled myosin V (red) and myosin VI (blue) movement on an actin filament (green). The net compliance in the coupled system is modeled as a simple harmonic spring with stiffness $k_s$. Each motor takes either a forward or backward step, based on whether the inter-motor tension after the step ($T$) is below or above the stall force ($F_{low}$ — myosin VI; $F_{high}$ — myosin V). (B) Outward flux of the mixed-motor ensemble ($\Phi_{out}$) on single actin filaments as a function of the normalized inter-motor tension per step ($\Delta T/F_{low}$) and stall force ratio ($r_s = F_{high}/F_{low}$). Based on previously reported stall forces for myosin V (*Mehta et al., 1999*; *Uemura et al., 2004*) and VI (*Rock et al., 2001*; *Nishikawa et al., 2002*; *Altman et al., 2004*), $r_s = 1.5$ and is indicated by the gray shaded region (*left*). The corresponding experimentally measured $\Phi_{out}$ (⊗; *Figure 2*) and $r_s = 1.5$ yield a $\Delta T/F_{low} = 0.55 \pm 0.01$. (C) Schematic forward step of a myosin with flexible (*left*) or rigid (*right*) lever arm on a digitized keratocyte actin network (green). The motor domains of the stepping motor (light blue shoes), non-stepping motor (gray shoes), lever arms, inter-motor linkage (pre-step—black spring; post-step—orange spring), and digitized actin network are drawn approximately to scale. The forward step results in an increase in both the inter-motor tension ($\Delta T \propto k_s$) and the intra-motor torsion ($\tau \propto k_F$). A flexible forward stepping motor ($k_F/k_s \ll 1$) minimizes inter-motor tension ($\Delta T_{low}$). A rigid forward stepping motor ($k_F/k_s \gg 1$) minimizes intra-motor torsion ($\Delta T_{high}$). (D) Simulated $\Delta T$ as a function of $k_F/k_s$. Varying lever arm rigidity ($k_F/k_s$) is sufficient to modulate $\Delta T$. (E) Outward flux of the mixed-motor ensemble ($\Phi_{out}$) on the keratocyte actin network as a function of the relative tension per step of the two motors ($\Delta T_{high}/\Delta T_{low}$). Gray shaded region (*left*) indicates the parameter space for $\Delta T/F_{low} = 0.55 \pm 0.01$ (see B). The corresponding experimentally measured $\Phi_{out}$ (⊕; *Figures 2, 3*) yields a $\Delta T_{high}/\Delta T_{low} = 1.20 \pm 0.05$. This enhanced $\Delta T$ for rigid motors evens out the competition on a branched 2D network compared to single filament tracks.

The following figure supplements are available for figure 5:

**Figure supplement 1**. Description of stochastic simulation.

**Figure supplement 2**. Actin network pore size alters tension generated.

**Figure supplement 3**. Inter-motor stiffness influences inter-motor tension.

motor (↓$\Phi_{out}$ with ↑$\Delta T_{high}/\Delta T_{low}$; *Figure 5E*). The experimentally measured $\Phi_{out}$ on 2D networks is significantly lower than on single actin filaments (⊗ vs ⊕; *Figure 5E*). Based on this measurement, the simulations estimate a $\Delta T_{high}/\Delta T_{low} = 1.20 \pm 0.05$ (⊕; *Figure 5E*) that is consistent with a higher flexural rigidity for myosin V (*Hariadi et al., 2014*) (*Figure 5D*). Therefore, our simulations reveal that the greater flexural rigidity of myosin V compared to myosin VI is sufficient to equalize the competition on 2D networks.

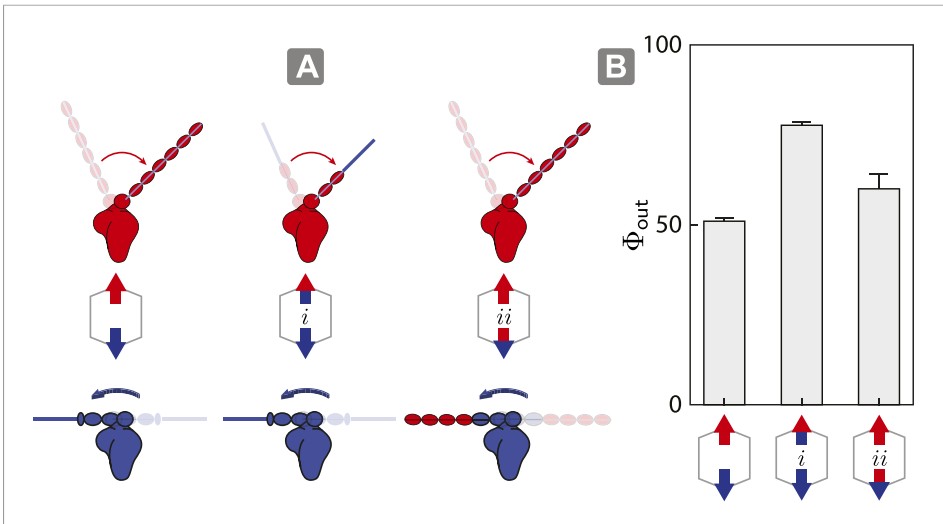

**Figure 6**. Switching lever arms restores myosin V dominance. (**A**) Scaffold and motor schematics used in the lever arm competition experiments. Lever arm rigidity was balanced by engineering the myosin V motor domain with the flexible lever arm of myosin VI (*i*; flexible vs flexible competition), or the myosin VI motor domain chimera with the rigid lever arm of myosin V (*ii*; rigid vs rigid competition). Arrowheads and arrowtails depict the myosin heads and lever arms, respectively (red—myosin V; blue—myosin VI). (**B**) Outward flux ($\Phi_{out}$) of indicated motor ensembles. Error bars are S.E.M. and were generated by bootstrapping ($N \geq 126$ trajectories; $\geq 3$ keratocytes).

## Swapping lever arms restores the dominance of the myosin with a higher stall force

As described above, the model shows that the outward flux of scaffolds composed of opposing motors on actin is influenced by the interplay between inter-motor tension and intra-motor strain (*Figure 5*). To test this model, we engineered a myosin V/VI chimera containing the myosin V motor domain with the flexible myosin VI lever arm and a myosin VI/V chimera consisting of the myosin VI motor domain with a rigid myosin V lever arm. These chimeras allow us to assess the competition involving opposing motors with similar lever arm rigidity, specifically 1V/VI:1VI (*Figure 6A* (*i*); flexible vs flexible) and 1VI/V:1V (*Figure 6A* (*ii*); rigid vs rigid). In both cases, the flexural rigidities and the changes in tension per step ($\Delta T$) of the opposing motors were estimated to be similar. The outward flux for ensembles involving either chimera (*Figure 6A*; *i* or *ii*) on 2D actin networks are significantly higher than scaffolds with 1V:1VI (*Figure 6B*). The higher outward flux indicates that balancing the tension per step ($\Delta T$) between the antagonistic motors is sufficient the restore the dominance of the stronger motor (myosin V) on 2D actin networks.

## Discussion

Myosin V and VI are antagonistic motors that cohabit membrane vesicles in neuronal growth cones (*Suter et al., 2000*). Myosin V is implicated in secretory traffic, whereas myosin VI is important for timely endocytosis (*Suter et al., 2000*; *Sahlender et al., 2005*; *Kneussel and Wagner, 2013*). While previous studies have examined the competition between myosin V and VI (*Ali et al., 2011*; *Hariadi et al., 2014*), a systematic measurement of their collective behavior is unexplored and formed the focus of this study. We report that while antagonistic motor ensembles display solely unidirectional movement, as previously reported (*Ali et al., 2011*; *Hariadi et al., 2014*), their directional sorting can be linearly tuned by the relative number of the two motor types. Further, this directional sorting can be modulated independently by the myosin lever arm and the actin architecture. These observations support a simple generalizable model, wherein competitive outcome is dependent on the ratio of the stall forces of the antagonistic motor types. Taken together, our findings provide an elegant mechanism for regulating vesicle sorting mediated by unconventional myosins, without the need to either segregate motor subtypes to distinct cargo (*Hartman et al., 2011*) or engage regulatory proteins that preferentially modulate the accessibility or activity of one of the motor types (*Fu and Holzbaur, 2014*).

One model for the observed unidirectional transport is that only a single myosin on the DNA scaffold interacts with the actin network at any given time. Under these conditions, the probability of movement towards the cell center or periphery is proportional to the relative number of myosin VI or V, respectively. However, this mechanism for the observed sorting is refuted by four distinct observations. First, as previously reported (*Hariadi et al., 2014*), scaffold run length linearly increases with motor number (*Supplementary file 2*), suggesting that multiple motors on the same scaffold are capable of interacting with the actin network. Second, scaffold speed changes substantially with varying number and type of myosin motors (*Figure 3*). For scaffolds with a given number of myosins, those with a single subtype move faster than those with both myosin V and VI. The reduction in speed with both myosin subtypes is consistent with the trailing myosin interacting with the actin network, despite the unidirectional movement of the scaffold. This interpretation is consistent with the coordinated back-stepping of the trailing motor observed by *Ali et al. (2007)*. Third, for scaffolds with both subtypes the speed decreases as the numbers of myosin V and VI are more evenly matched. This systematic reduction in speed alone argues for progressive engagement of additional antagonistic motors with the actin network. Fourth, experiments with photo-cleavable myosin-scaffold linkages further show that removal of the antagonistic myosin leads to either an increase in speed (release trailing motor) or a reversal in direction (release leading motor) (*Figure 4*). Altogether, given these observations we propose an alternate model that emphasizes inter-motor interactions (*Figure 5*).

Our model identifies the ratio of the stall forces ($r_s = F_{high}/F_{low}$) of the antagonistic motors as the key parameter that determines the outcome of scaffold sorting on 1D or 2D actin filament tracks. For ensembles with mismatched motors ($r_s > 2$), the model predicts unidirectional movement led solely by the stronger motor. However, for motors of similar strength ($1 < r_s < 2$), the model predicts that either motor may lead the unidirectional motion, with the directional flux of scaffolds dependent on $r_s$. These predictions are consistent with the unidirectional movement observed here, and in a previous report based on experiments using a single myosin V and a single myosin VI attached to a quantum dot (*Ali et al., 2011*). Further, the model is in agreement with a recent report of unidirectional trajectories for DNA scaffolds linked to both kinesins and dyneins on 1D microtubules (*Derr et al., 2012*; *Roberts et al., 2014*). However, it differs from the bi-directional movement of isolated endosomes that are driven by a combination of native kinesin and dynein motors (*Soppina et al., 2009*). We speculate, though, that bi-directional movement stems from the influence of additional regulatory elements on native endosomes (*Kunwar et al., 2011*). Lastly, our model explains the differences in sorting observed in actin filaments vs 2D networks, and the role of the myosin lever arm in regulating sorting. In essence, motors with greater intra-motor torsional strain (rigid lever) experience a larger inter-motor tension per step and hence lose their competitive edge on 2D networks.

The lever arm of myosin is primarily regarded as a mechanical amplifier in its chemo–mechanical cycle (*Spink et al., 2008*). Our study, however, suggests a broader regulatory role for the lever arm in membrane trafficking. We find that the structural properties of the myosin lever arm control the directional flux of scaffolds on our model cellular actin network, thus having implications on sorting of vesicular cargo. Beyond this observation, structural elements in myosins have been shown to influence motility on actin networks. For instance, an extension of the myosin X lever arm is necessary for its preferential processive movement on parallel actin bundles, but not on single actin filaments (*Brawley and Rock, 2009*; *Nagy and Rock, 2010*). Myosin VI, on the other hand, has a unique three-helix bundle in its lever arm, which can unfold to alter the motor's structural properties (*Mukherjea et al., 2014*). Furthermore, for groups of myosin V and VI, the flexibility of the lever arm controls trajectory shapes on 2D actin networks (*Hariadi et al., 2014*). In addition to myosin structure, actin architecture also influences myosin function. For example, single myosin V and VI have different stepping dynamics at actin filament intersections (*Ali et al., 2007*) then on actin bundles (*Ali et al., 2013*). An in situ motility assay using detergent-extracted cells also reported that individual myosin V, VI, and X dimers show preferential motility on different actin architectures (*Brawley and Rock, 2009*). Together, these studies suggest a subtler regulation of cellular processes that emerges from unique structural features in myosins that modulate either individual or collective actin-myosin interactions.

# Materials and methods

## Buffer and reagents

1× Assay Buffer (AB Buffer): 25 mM imidazole (pH 7.5), 4 mM MgCl$_2$, 1 mM EGTA, 25 mM KCl, 1 mM DTT; 1× AB.BSA: AB buffer + 1 mg/ml BSA; 1× AB.BSA.CAM: AB.BSA buffer + 9 µM calmodulin.

## Preparation of Benzyl-guanine-labeled oligonucleotide

Benzyl-guanine NHS ester (BG-GLA-NHS; NEB, Ipswich, MA) was covalently linked to the C6-amine modified oligonucleotides (BG-oligo 1 and BG-oligo 5; *Supplementary file 1*). Briefly, 0.17 mM C6-amine-oligo-Cy3 was incubated with 11.6 mM BG-GLA-NHS in 0.1 M NaBO$_3$ for 2–4 hr at 37°C with shaking. BG-labeled oligo was purified twice through Illustra G-50 micro columns (GE Healthcare, Pittsburgh, PA) equilibrated in 2 mM Tris, pH 8.5. BG-oligo concentration was determined from absorbance at 260 nm.

### Myosin preparation and labeling

Myosin V, VI, V/VI, and VI/V were constructed, expressed in Sf9 insect cells, purified, and oligo-labeled as previously described (*Hariadi et al., 2014*). Constructs contained from N- to C-terminus, myosin motor domain and lever arm, a GCN4 leucine zipper (for dimerization), SNAP tag, a FLAG tag (for purification), and finally a 6xHis tag (alternative purification tag). Myosin VI contained residues 1–992 from *Sus scrofa* and myosin Va, residues 1–1103 from *Gallus gallus.* For V/VI, residues 1–815 of myosin V were followed by the lever arm of myosin VI (residues 917–992). For VI/V, residues 1–810 of myosin VI were followed by residues 767–1103 of myosin V. Myosin VI and V/VI were cloned in pBiex-1 (EMD Millipore, Germany) and expressed through transient transfection using the Escort VI system (Sigma, St. Louis, MO). Myosin V and VI/V were cloned in pFastBac for calmodulin co-expression and expressed through baculovirus infection. Cells were lysed, incubated with Anti-FLAG resin (Sigma), and washed according to *Hariadi et al. (2014)*. Myosin bound to Anti-FLAG resin was incubated with excess (>5 µM) BG-oligo-Cy3 at 37°C for 30 min followed by overnight incubation on ice. Resin was washed three times with Wash Buffer (20 mM Imidazole, 150 mM KCl, 5 mM MgCl$_2$, 1 mM EDTA, 1 mM EGTA, 1 mM DTT, 1 µg/ml PMSF, 10 µg/ml aprotinin, 10 µg/ml leupeptin, pH 7.4). Resin was then washed twice with Wash Buffer + 55% (vol/vol) glycerol. Finally, BG-oligo-labeled myosin was incubated with 0.2 mg/ml FLAG-peptide (Sigma). Calmodulin was added to 9 µM and protein was stored at −20°C. Labeling efficiency was assessed with a 10% SDS-PAGE gel as labeled myosin displayed a distinct gel-shift.

## Scaffold-myosin preparation

DNA nanostructures were prepared based on the detailed description in our previous work (*Hariadi et al., 2014*). The sequences for the scaffold and all oligonucleotides are listed in *Supplementary file 1*. Each origami scaffold is labeled with 23 Cy3 molecules (*Figure 1—figure supplements 1, 2*; *Figure 3—figure supplement 1*; *Supplementary file 1*) for high signal-to-noise imaging and contains a biotinylated-strand to facilitate removal of unbound myosins. Single-stranded M13mp18 DNA (NEB) were mixed with fourfold excess of short stable strands (IDT, Coralville, IA), followed by 2 hr annealing as previously described (*Rothemund, 2006*). Intact scaffolds were separated from excess staple strands using Amicon Ultra 100K cutoff spin columns (EMD Millipore). Purified scaffolds were mixed with excess labeled myosin, a mixture of 42-nt oligos with randomized sequences (blocking oligos), and 1–5 µM calmodulin in 1× AB.BSA. After 10 min of incubation at room temperature, excess streptavidin-coated magnetic beads (NEB) were added and incubated at room temperature with shaking for 10 min. The beads were washed with AB.BSA.CAM. Finally, the beads were incubated in AB.BSA.CAM containing an imaging solution of 2 mM ATP, 1 mM phosphocreatine, 0.1 mg/ml creatine-phosphokinase, 45 µg/ml catalase, 25 µg/ml glucose oxidase, 1–2% glucose, and excess elution strand for strand displacement of origami from streptavidin magnetic beads.

### Single actin filament assay

Motility assays were acquired at 120× magnification on an objective-based TIRF microscope (Olympus IX81) with a 60× NA 1.48 Apo TIRF objective (Olympus), 2× image magnifier, EMCCD iXON Ultra, and a 488 nm laser (CUBE 488–50, for actin filaments), a 532 nm laser (CrystaLaser, CL532-150 mW-L, for excess Cy3-labeled myosin VI) and a 640 nm laser (Coherent, CUBE 640-100, for Cy5-labeled DNA scaffold). Motility assays were performed using plasma-cleaned rectangular

capillary tubes (EMS, 75 mm × 50 µm × 1 µm). First, biotinylated, 488Alexa-phalloidin-stabilized actin filaments were immobilized to the inner surface of the capillary tube by BSA-biotin-neutravidin-linkages. Unbound actin filaments were washed with AB.BSA. Purified myosin-scaffold complexes in AB.BSA. CAM + imaging reagents (2 mM ATP, 1 mM phosphocreatine, 0.1 mg/ml creatine-phospho-kinase, 25 µg/ml glucose-oxidase, 45 µg/ml catalase, 1% glucose, 1 µM random library 42-nt ssDNA) were added to the capillary chamber. For each field of view, the polarities of the Alexa488-phalloidin stabilized actin filaments were determined from a 1 min motility movie of the remaining unbound Cy3-labeled myosin VI in solution. The purification step was estimated to remove >95% of excess myosin motors. Movies of Cy5-scaffold motility on the actin filaments were obtained at 2 Hz for ≥30 min.

## Keratocyte assay

Keratocytes were derived from scales of *Thorichthys meeki* (Firemouth Cichlids) or *Rocio octofasciata* (Jack Dempsey Cichlids) as previously described (*Sivaramakrishnan and Spudich, 2009*). All protocols conform to the guidelines of the local animal care and use committee (IACUC). Extracted keratocytes were washed with AB.BSA and stabilized with phalloidin (50 nM Alexa-488 phalloidin [Invitrogen] and 200 nM unlabeled phalloidin [Sigma]). Purified Cy5-labeled origami-myosin scaffolds in AB.BSA. CAM buffer containing imaging reagents (2 mM ATP, 1 mM phosphocreatine, 0.1 mg/ml creatine-phospho-kinase, 25 µg/ml glucose-oxidase, 45 µg/ml catalase, 1% glucose, 1 µM random library 42-nt ssDNA) were added to extracted keratocytes as previously described (*Hariadi et al., 2014*). Time-lapse imaging was taken using 150× magnification on a Nikon TiE microscope equipped with a 100 × 1.4 NA Plan-Apo oil-immersion objective, 1.5 magnifier, a mercury arc lamp, Evolve EMCCD camera (512 pixel × 512 pixel; Photometrics), Nikon Perfect Focus System, and Nikon NIS-Elements software.

## Photo-cleavable myosin experiments

Myosin-scaffolds were composed of either three photo-cleavable myosin V and three myosin VI or three myosin V and three photo-cleavable myosin VI (*Figure 4A–B*). Removal of one myosin type from the scaffold was achieved by introducing a photocleavable element in selected myosin-attachment DNA linkers (*Supplementary file 1*). Myosins with photo-cleavable DNA linkers were released from the scaffolds through continuous excitation with UV laser (405 nm; Coherent CUBE 405–100; 0.2 mW exposure). The trajectories were classified into four classes of movement based on the position of the photo-cleavable linkers and pre/post events (*Figure 4*).

## Data analysis

Trajectories of individual scaffold-myosin complexes were analyzed using custom MATLAB Particle Tracking software (*Churchman et al., 2005*) and Imaris (Bitplane). A 2D-gaussian fit was used to estimate scaffold position with sub-pixel resolution. Intensity of scaffold was used to exclude scaffold dimers. Directional flux analysis was automated with custom-code in Mathematica (available at https://github.com/rizalhariadi/DirectionalAnalysis). Analysis of directional scaffold movement was performed only to scaffold-myosin complexes that moved for more than 6 continuous frames (3 s) and covered a distance of more than 8 pixels (860 nm). For experiments on keratocytes, the directional trajectory was determined as follow. First, a local actin polarity vector was calculated for each trajectory (*Hariadi et al., 2014*). Then, the movement direction for each trajectory was the calculated by comparing two Euclidian distances along the actin polarity field vector; (1) distance between starting point and cell periphery, $\Delta x_{start}$, and (2) distance between finish point and cell periphery, $\Delta x_{finish}$. Trajectories away from the cell center have a negative ($\Delta x_{finish} - \Delta x_{start}$), whereas trajectories toward the cell-center are positive.

## Simulation of competitive movement

1. The centers of mass of the two motors are connected by a simple linear spring of stiffness $k_s$ and initial inter-motor tension $T = 0$.
2. The stall force for myosin V and VI are defined as $F_{high}$ and $F_{low}$, respectively.
3. The dwell time is assumed to be exponentially distributed, and a series of discrete dwell times are derived from previously measured mean dwell times for myosin V (170 ms) and VI (215 ms) (*Hariadi et al., 2014*).
4. The motor with the shorter dwell time steps first.
5. A myosin step increases $T$.
   a. For movement on an actin filament, a myosin step increases the tension by $\mathcal{N}(\Delta T, 0.1\,\Delta T)$ regardless of motor type, where $\mathcal{N}(\mu, \sigma)$ represents a normal distribution with mean = $\mu$ and standard deviation = $\sigma$.
   b. For movement on keratocytes, a myosin V step increases the tension by $\mathcal{N}(\Delta T_{high}, 0.1\,\Delta T_{high})$ whereas a myosin VI step increases the tension by $\mathcal{N}(\Delta T_{low}, 0.1\,\Delta T_{low})$.

6. After each motor step, the resulting $T$ is used to modify the dwell time of each motor as follows:
   a. A linear force-speed relationship is assumed for both motors.
   b. $T$ is used to calculate the mean speed ($\nu$) and mean dwell time ($\propto \nu^{-1}$) for each motor. The load-dependent speed is given by $v = v_0 (1 - T/F_{stall})$, where $\nu_o$ is the zero-strained speed, $F_{stall}$ is the stall force of the stepping motor ($F_{high}$ or $F_{low}$) and $T$ is the inter-motor tension ($0 \leq T \leq F_{stall}$).
   c. The discrete dwell time distribution of each motor is modified in proportion to the estimated mean dwell time after each step.
7. The new discrete dwell time distribution is used to identify the next stepping motor, with a repeat of steps 5-7.
8. If $T$ is larger than the stall force of the stepping motor ($F_{high}$ or $F_{low}$), then this motor undergoes a conformational change that leads to preferential back-stepping (Gebhardt et al., 2006). This motor is designated as the 'losing' motor. The scaffold is now primed for unidirectional movement lead by the 'winning motor'.
9. Steps 1–8 are simulated over ≥1000 times for each value of $\Delta T$, $r_s$ ($F_{high}/F_{low}$), and for simulations on keratocytes a given value of ($\Delta T_{high}/\Delta T_{low}$). For each simulation:
   a. If the stronger motor ($F_{high}$) is the winning motor $n_+ = n_+ + 1$.
   b. If the weaker motor ($F_{low}$) is the winning motor $n_- = n_- + 1$.
10. For each condition outward flux is calculated as $\Phi_{out} = n_+/(n_+ + n_-)$.
11. $\Phi_{out}$ measurements are plotted over a range of normalized $\Delta T$ ($\Delta T/F_{low}$), $r_s$, and ($\Delta T_{high}/\Delta T_{low}$).

Estimate inter-molecular tension per step ($\Delta T$) for movement on keratocytes as a function of $k_F/k_s$ (*Figure 5C–D* and *Figure 5—figure supplements 1–3*)—Stochastic simulations of movement of motor ensembles with lever arm flexural rigidity $k_F$ and inter-motor stiffness $k_s$ on the digitized actin network were performed in Mathematica (*Figure 5—figure supplement 1*). For movement on keratocytes, $\Delta T$ after each step is given by $k_s \bullet (\Delta x_{post} - \Delta x_{pre}) \leq k_s \bullet s$, where $\Delta x_{post}$ and $\Delta x_{pre}$ are the inter-motor extensions before and after the step and $s$ is the myosin step size on the actin filament. Note that movement on keratocytes gives rise to lower $\Delta T$ than the collective movement on single actin filament ($\Delta T = k_s \bullet s$). Mean and standard deviation in $\Delta T$ for each $k_F/k_s$ were computed from 400 simulated steps (*Figure 5D*).

## Statistical analysis

### Bootstrap error estimates

Bootstrapping was used to estimate the uncertainty (S.E.M.) of a measurement $X$ (*Figures 2–4, 6*). From the data set of size N for measurement $X$, a subset of data points was randomly selected of size $\lfloor N/2 \rfloor$, where the brackets denote rounding off to the nearest integer. This was repeated ≥1000 times, and in each subset an element was never chosen more than once. An average measurement, $x_j$, was then generated for each data set. Finally, the standard deviation of $x_j$ was calculated from these measurements and used to estimate of the uncertainty in the measurement of $X$.

## Acknowledgements

The authors thank M Ritt and M Cale for technical assistance. Research was funded by the American Heart Association Scientist Development Grant (13SDG14270009) and the NIH (1DP2 CA186752-01 and 1-R01-GM-105646-01-A1). RFS is a Life Sciences Research Foundation Fellow.

# Additional information

## Funding

| Funder | Grant reference | Author |
| --- | --- | --- |
| National Institutes of Health (NIH) | 1-R01-GM-105646-01-A1 | Sivaraj Sivaramakrishnan |
| American Heart Association (AHA) | 13SDG14270009 | Sivaraj Sivaramakrishnan |
| National Institutes of Health (NIH) | 1DP2 CA186752-01 | Sivaraj Sivaramakrishnan |
| Life Sciences Research Foundation | postdoctoral fellowship | Ruth F Sommese |

The funders had no role in study design, data collection and interpretation, or the decision to submit the work for publication.

## Author contributions

RFH, Conception and design, Acquisition of data, Analysis and interpretation of data, Drafting or revising the article; RFS, Acquisition of data, Analysis and interpretation of data, Drafting or revising the article; SS, Conception and design, Analysis and interpretation of data, Drafting or revising the article

# Additional files

### Supplementary files

• Supplementary file 1. Computer aided staple strand sequences for the flat-rectangular DNA origami scaffold.

• Supplementary file 2. Summary of run length measurements.

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
