## [Decision Letter]

Thank you for sending your work entitled “Tuning myosin-driven sorting on cellular actin networks” for consideration at *eLife*. Your article has been favorably evaluated by Vivek Malhotra (Senior editor), a Reviewing editor, and 3 reviewers.

The Reviewing editor and the reviewers discussed their comments before we reached this decision, and the Reviewing editor has assembled the following comments to help you prepare a revised submission.

1) All reviewers found it difficult to evaluate this paper without reference to your recent 2014 PNAS paper. In some aspects (e.g., the flexural rigidity of the motor dimers) the new paper is almost supplementary to the PNAS paper. But in other aspects (speed and direction of movement, as opposed to trajectory) the new paper makes original points. The paper also has a compelling final model that simply explains the results over various conditions of the study yet assumes only a limited number of motor properties (ratio of motor stall forces, stepping rates, stiffness of intermotor linkage and motor lever arm stiffness). The reviewers request that you more clearly distinguish the new results from the previous ones in the Introduction and Discussion, rewriting as necessary. For example, in the last paragraph of the Introduction claim 3 findings: point 3 was clearly made in the PNAS paper and point 1 was partly addressed there before.

2) Please address the substantive concerns of Reviewers 1 and 2 below. It does not appear that any of these concerns will need new experiments.

*Reviewer #1*:

1) In the subsection headed “Actin architecture influences cargo sorting of antagonistic motor assemblies”, why do results differ from previous results? Please explain.

2) In the same section: the track persistence is an important point that is not discussed. I understand that it discussed in the PNAS paper, but not in the same conditions of relative motor number, and it is striking from Figure 3 that the tracks are sometimes very short depending on the number of motors. The absolute velocities should be given in the text at that point, compared with single motor velocities, and discussed. It seems from Figure 2 that two same motors together move 3 times slower than a single one. Can this be explained? The number of different experiments should be indicated in the figure legend.

3) Model: the yellow spring Figure 1 could be better defined. As I understand it is the spring constant of the DNA scaffold? Or is it the one of the motor tail? By the way, how much do we expect? Are we only interested in the relative value of this spring constant with the flexural rigidity? The flexural rigidity k_F_ is not clearly defined (“as previously described [in the PNAS paper]”, in the subsection “Stochastic simulations identify key parameters that drive unidirectional movement”), whereas the spring constant k_s_, which is very classical, is well defined in Figure 5. More effort should be made in making clear to the author what k_F_ exactly represents (although, again, this can be found in the PNAS paper).

4) Simulations: I understand that simulations are done in 1D? One question that arises is how the structure of the actin network could influence the results? How can one myosin jump on an Arp2/3 branch? Does the length of the trajectory correspond to a single branch? These questions could be addressed by a 2D simulation. Along the same line, “both motors take forward steps stochastically in opposite directions” (in the subsection “Stochastic simulations identify key parameters that drive unidirectional movement”); is that true also on 2D networks?

5) In Figure 5, it is not clear why PHI=90% when deltaT=0. Please explain.

6) In the same subsection, could the network structure explain that PHI is significantly lower than on single filaments? Could the authors compare track persistence between single filaments and 2D networks? Can the authors comment on step size compared to network mesh size?

7) In the beginning of the Discussion, it is very important that motor assemblies are in steady state motion. Since the authors have simulations, I would expect some explanation. What would change directionality? Can the authors propose a way of testing it experimentally?

*Reviewer #2*:

1) In the subsection “Actin architecture influences cargo sorting of antagonistic motor assemblies”, the authors mention that the plus-end biased flux differs from their findings in Hariadi et al., PNAS, 2014; why?

2) What is the physiologically relevant actin network architecture on which transport takes place? If it's the actin cortex, which is a tight meshwork of isotropically oriented filaments, then how well is it really modeled by the branched architecture of actin in the lamellipodium?

---

## [Author Response]

*1) All reviewers found it difficult to evaluate this paper without reference to your recent 2014 PNAS paper. In some aspects (e.g., the flexural rigidity of the motor dimers) the new paper is almost supplementary to the PNAS paper. But in other aspects (speed and direction of movement, as opposed to trajectory) the new paper makes original points. The paper also has a compelling final model that simply explains the results over various conditions of the study yet assumes only a limited number of motor properties (ratio of motor stall forces, stepping rates, stiffness of intermotor linkage and motor lever arm stiffness). The reviewers request that you more clearly distinguish the new results from the previous ones in the Introduction and Discussion, rewriting as necessary. For example, in the last paragraph of the Introduction claim 3 findings: point 3 was clearly made in the PNAS paper and point 1 was partly addressed there before*.

The Abstract, Introduction, Results, and Discussion have been substantially re-written with the specific intent to highlight the novel findings of this study as differentiated from our previous PNAS manuscript.

a) The Abstract is now limited to the novel findings of this study, without overlap with findings presented in our PNAS paper.

b) Paragraph 3 of the Introduction summarizes previous findings, including those from our lab (Hariadi et al., PNAS 2014) and the Warshaw lab *(*Ali et al., PNAS 2011), and concludes with outstanding questions related to competitive outcome of antagonistic myosin ensembles that motivated the current study.

c) Paragraph 4 of the Introduction now summarizes the novel findings of the current study that are distinct from our previous study. The summary comments of the reviewing editor concur with the significance of these findings.

d) Paragraph 2 of the Results has now been re-written to distinguish previous observations from those being made in the current study. In this section, we emphasize that we can now resolve a previous discrepancy in the competitive outcome between Hariadi et al*.* (PNAS 2014) and Ali et al*.* (PNAS 2011).

e) The first paragraph of the Discussion has now been re-written to highlight the key contributions of the current study. The focus of this manuscript is the mechanisms underlying the competitive outcome in antagonistic motor ensembles and their impact on the collective function of myosins in membrane sorting.

*2) Please address the Substantive Concerns of Reviewers 1 and 2 below. It does not appear that any of these concerns will need new experiments*.

We concur that reviewer comments do not motivate additional experiments. Individual responses to the referees’ comments are shown below:

Reviewer #1:

*1) In the subsection headed “Actin architecture influences cargo sorting of antagonistic motor assemblies”, why do results differ from previous results? Please explain*.

We apologize for the confusion in the organization of the initial sections of the manuscript (see response (a)) to the Reviewing Editor). We have now extensively re-written the second section of the Results to delineate the contributions of scaffold type (quantum dot versus DNA nanostructure), motor number, and actin architecture. Further, we have clearly stated in the Introduction (paragraphs 3 and 4) the information derived from previous studies, the questions left unanswered, and subsequently the results of the current study.

*2) In the same section: the track persistence is an important point that is not discussed. I understand that it discussed in the PNAS paper, but not in the same conditions of relative motor number*,

The run length was not a significant new finding in this manuscript. However, we agree that this is an important aspect that needs to be delineated. We have now appended a supplementary table of the mean run lengths for all experimental conditions (*x*V:*y*VI; [Supplementary-material SD2-data]).

*And it is striking from*
Figure 3
*that the tracks are sometimes very short depending on the number of motors*.

As the reviewer astutely points out, some of the trajectories in Figure 3 are very short. We expect that the run length for each *x*V:*y*VI competition condition to follow an exponential distribution with PDF(x) = λ^–1^ e^–x/λ^, where λ is the mean run length. As a consequence, the probability of observing very short trajectories is not negligible. As an example, for a run length distribution with mean λ = 1000 nm, the probability to observe a trajectory that is shorter than 300 nm is CDF(*x* = 300 nm) = 1–e^–x/λ^ = 1–e^–(300 nm)/(1000 nm)^ = 26%.

The relatively short mean run lengths are consistent with the previously published mean run lengths for 1V:1VI competition experiments along 1D actin filaments, albeit with quantum dot scaffolds (Ali et al.*,* PNAS, 2011). In that study, the mean run lengths of the plus-end and negative-end directed movements of 1V:1VI scaffolds on 1D actin filaments were measured to be 0.54 µm and 0.63 µm, respectively. Ali et al.'s measurements are similar to our results for the movement 1V:1VI scaffolds ([Supplementary-material SD2-data]) along 1D filaments (0.59–0.77 µm) and on 2D actin networks (0.63–0.64 µm).

*The absolute velocities should be given in the text at that point, compared with single motor velocities, and discussed*.

We have added the absolute speed in the main text and compared the measured mixed ensemble speeds with single motor type ensemble speeds (paragraph 4 of the Results section).

*It seems from*
Figure 2
*that two same motors together move 3 times slower than a single one*. *Can this be explained?*

Multi-myosin V and VI ensembles have been shown to move at significantly slower speeds compared to single myosins (Sivaramakrishnan and Spudich, JCB, 2009, Lu et al., JBC, 2012). We do observe a significant drop in the speed of the ensemble compared to a single one, although a lesser extent than suggested by the reviewer. In Figure 2 the two same motors move 28% (2V:ØVI scaffolds) or 11% (ØV:2VI scaffolds) slower than single myosin V or VI, respectively. The large step sizes and relatively low stall forces of myosin V or VI lead to the reduction speed in groups of myosin motors (Lu et al., JBC 2012).

*The number of different experiments should be indicated in the figure legend*.

We have modified the figure legends of Figures 2, 3, 4 and 6 to indicate the number of independent keratocyte samples that were analyzed for each experimental condition.

*3) Model: the yellow spring*
Figure 1
*could be better defined*. *As I understand it is the spring constant of the DNA scaffold? Or is it the one of the motor tail?*

We have now included a better definition of the inter-motor tension in the modeling section (paragraph 5 of the Results section). In brief, *k*_*s*_ is the effective spring constant of the motor complex. For two motor assemblies the effective spring constant can be modeled as three springs in series, *k*_*s*_^–1^ = *k*_*myosin1*_^–1^ + *k*_*DNA*_^–1^ + *k*_*myosin2*_^–1^, where *k*_*DNA*_ and *k*_*myosin*_ are the stiffness of the DNA nano-structure and myosin motors, respectively. Here, we assume that both myosin V and VI have similar stiffness under tension. Since DNA is more rigid polymer than the myosin molecule (*k*_*DNA*_>>*k*_*myosin*_), the previous equation reduces to *k*_*s*_ ∼ *k*_*myosin*_ / 2.

*By the way*, *how much do we expect?*

The stiffness of dimeric myosin V has been measured to be 0.18 – 0.35 pN/nm (Veigel et al., Nat Cell Bio 2001), which leads to *k*_*s*_ ∼ 0.09 – 0.18 pN/nm. We can also infer *k*_*s*_ from our model in the Figure 5. For stall force ratio *r*_*s*_ = *F*_*high*_
*/ F*_*low*_
*=* 1.5 and experimentally measured flux Φ_out_ = 68%, the simulation gives *∆T* = 0.55 *F*_*low*_. Using the stall force of myosin VI (*F*_*low*_ ∼ 2 pN) and the step size of myosin VI (s ∼30 nm), the effective spring constant can be estimated to be *k*_*s*_ = *∆T / s* = 0.55 *F*_*low*_
*/ s* = 0.55 × 2 pN */* (30 nm) ∼ 0.04 pN/nm, which is within a factor of 3 of the value from Veigel et al.

*Are we only interested in the relative value of this spring constant with the flexural rigidity*?

The reviewer raises an important concern. It was unclear in our initial submission if ∆*T* is sensitive to the absolute value of *k*_*F*_ and *k*_*s*_ or just their ratio (*k*_*F*_
*/ k*_*s*_). To answer the referee's question, we ran additional simulations of collective movement of one motor type with different *k*_*F*_'s and *k*_*s*_'s on 2D networks (Figure 5—figure supplement 3). These simulations predict that the ∆*T* is sensitive to the absolute value of *k*_*F*_ and *k*_*s*_ in addition to their relative value. Therefore, in the revised manuscript, we plotted the directional flux as a function of *∆T*_*high*_
*/ ∆T*_*low*_, instead of *k*_*F*_
*/ k*_*s*_ or *k*_*F,high*_
*/ k*_*F,low*_.

*The flexural rigidity k*_*F*_
*is not clearly defined (*“*as previously described [in the PNAS paper]*”*, in the subsection* “*Stochastic simulations identify key parameters that drive unidirectional movement*”*), whereas the spring constant k*_*s*_*, which is very classical, is well defined in*
Figure 5*. More effort should be made in making clear to the author what k*_*F*_
*exactly represents (although, again, this can be found in the PNAS paper)*.

We thank the referee for these suggestions. We have now added the description of the flexural rigidity *k*_*F*_ in the revised main text (paragraph 7 of the Results section and Figure 5—figure supplement 1, panel L).

*4) Simulations: I understand that simulations are done in 1D? One question that arises is how the structure of the actin network could influence the results? How can one myosin jump on an Arp2/3 branch? Does the length of the trajectory correspond to a single branch? These questions could be addressed by a 2D simulation. Along the same line,* “*both motors take forward steps stochastically in opposite directions*” *(in the subsection* “*Stochastic simulations identify key parameters that drive unidirectional movement*”*); is that true also on 2D networks*?

Yes, the simulations for both 1D and 2D conditions are along a 1D track. In the revision (paragraph 7 of the Results section) we now state this upfront along with the rationale for simulating both 1D and 2D conditions along a 1D track. Specifically, one major effect of stepping in 2D is that the inter-motor tension (∆*T*) is now dependent on the flexural rigidity (*k*_*F*_) of the forward stepping motor. Since we were primarily interested in the outcome of the competition, this parameter was incorporated in 1D simulations by varying (∆*T*) based on the relative flexural rigidity (*k*_*F*_
*/ k*_*s*_). It was not our intent to simulate either a single branch or capture features such as the Arp2/3 complex that might influence myosin movement. We sincerely apologize for the confusion, as it is evident that the original submission was unclear in its description of rationale and methodology. The reviewer does raise an interesting point regarding the influence of the structure of the actin network. To address this, we ran 2D simulations with actin networks of different mesh sizes (Figure 5—figure supplement 1 and Figure 5—figure supplement 2). While the actin network mesh size did alter the absolute values of ∆*T* for a given *k*_*F*_
*/ k*_*s*_, ∆*T* does significantly increase with *k*_*F*_
*/ k*_*s*_ supporting the underlying argument for approximating 2D conditions with a 1D track.

*5) In*
Figure 5*, it is not clear why PHI=90% when deltaT=0. Please explain*.

We are grateful that the referee caught this error in our simulation. As hinted by the referee, as ∆T approaches 0, all collective movements should be dominated by the stronger motor (Φ=100%). The erroneous value reported in the first submission stemmed from an error in our implementation of the software algorithm. We have thoroughly proof-read our software code, re-run the stochastic simulations, and replaced the incorrect plots with the latest simulation outputs in the Figure 5. Although the graphs in the Figure 5 do change as a result of this exercise, the conclusion of our model in the originally submitted paper is still valid. Because the change was made in the computer code, and not to the simulation algorithm, we did not make any revision to the simulation section in the Materials and methods. We will make the software code freely available upon request.

*6) In the same subsection, could the network structure explain that PHI is significantly lower than on single filaments*?

The reviewer’s point is well taken. It is possible that an asymmetry in the actin architecture (looking towards the cell center or the cell periphery) could influence Φ such that myosin VI has an advantage. However, our ability to reverse directional flux by altering the myosin lever arm (Figure 6) suggests otherwise. Specifically, providing myosin V with the flexible lever of myosin VI significantly biases the flux in favor of myosin V. In this experiment, the myosin still moves in the same direction but with an altered lever arm. Hence, while network structure could bias Φ relative to single actin filaments it does not appear to be a dominant effect.

*Could the authors compare track persistence between single filaments and 2D networks*?

To address this question, we quantified the run lengths for all *x*V:*y*VI scaffolds on 1D actin filaments and 2D keratocyte actin networks ([Supplementary-material SD2-data]). On both actin architectures, the measured run lengths are limited by the actin filament length (∼25 µm) or keratocyte width (∼5 µm). Scaffolds that reach the tip of an actin filament or keratocyte edge have to end their movement prematurely. Therefore, a direct comparison of mean run lengths in the 2 actin tracks is not very meaningful. Ignoring the finite actin effect, the mean run length for the +end and –end directed movement of 1V:1VI scaffolds along single filaments and on 2D networks are qualitatively similar ([Supplementary-material SD2-data]).

*Can the authors comment on step size compared to network mesh size*?

The mesh size (∼30 nm) is comparable to the step size of myosin V and VI along single actin filament (∼35 nm) and significantly smaller than the dimensions of the rectangular scaffolds (∼100 nm). The mean pore size of the keratocyte actin network has been measured to be ∼30 nm (37) as depicted in Figure 1. We have now added this information to the caption of Figure 1. The schematics are drawn approximately to scale.

*7) In the beginning of the Discussion, it is very important that motor assemblies are in steady state motion. Since the authors have simulations, I would expect some explanation*.

As stated earlier, all of our simulations are conducted along 1D tracks. The simulations continue until one of the motors win. Subsequently, for motors with matched stall forces (1< *r*_*s*_ < 2) the leading motor steps to increase the inter-motor tension while the trailing motor steps to decrease it. The trailing motor is not able to take an additional forward step in this scenario. This was the intent of our statement of a ‘quasi-steady state’. We realize that we do not have direct evidence besides our simulation to back up this statement. Hence, this sentence has now been removed from the main text of the manuscript. A complete description of the outcome of the simulation is now limited to the Methods section of the manuscript.

*What would change directionality? Can the authors propose a way of testing it experimentally*?

Our experiments involving photo-cleavable linkers (Figure 4) suggest that disengagement of the winning motor, even temporarily, is a simple way to change directionality. It is also plausible that an increase in the backwards force experienced by the leading motor, for instance as it moves through a viscoelastic network, can lead to its stalling. Under these circumstances, the competition will be biased towards the trailing motor. Specifically, optical tweezers can be used to temporarily apply a resistive load and ‘reset’ the competition between the motors. Also, regulatory events such as phosphorylation can be used to alter the stepping kinetics of one motor so as to bias the competition. Both of these scenarios are of interest and are a focus of future investigations in our lab.

Reviewer #2:

*1) In the subsection “Actin architecture influences cargo sorting of antagonistic motor assemblies”, the authors mention that the plus-end biased flux differs from their findings in Hariadi et al., PNAS, 2014; why*?

We apologize to the reviewer for the confusion concerning the findings of this study versus our previous work. We have now re-written the Introduction (paragraphs 3 and 4) to clearly state the information derived from previous studies and subsequently the results of the current study. Furthermore, the second section of the Results (paragraph 2) has been rewritten to clearly state the contributions of scaffold type, motor number, and actin architecture on ensemble competition and flux.

*2) What is the physiologically relevant actin network architecture on which transport takes place? If it's the actin cortex, which is a tight meshwork of isotropically oriented filaments, then how well is it really modeled by the branched architecture of actin in the lamellipodium*?

We agree with the intent of the referee’s suggestion, in that, the keratocyte lamellipodium is a model dense, branched, actin network that facilitates careful quantitative observations of the actin-myosin interactions in two-dimensions. We understand that our original submission implied otherwise. Accordingly, the Abstract, Introduction, and Discussion have been modified in accordance with the use of the keratocyte lamellipodium as a model actin network. Future studies on unconventional myosins will need to examine the influence of different actin architectures, and potentially 3D actin structures on the collective movement of unconventional myosins.